# Comparison of Selected Characteristics of SARS-CoV-2, SARS-CoV, and HCoV-NL63

**Darina Bačenková** [1,*] , **Marianna Trebuňová** [1], **Tatiana Špakovská** [2], **Marek Schnitzer** [1] , **Lucia Bednarčíková** [1] and **Jozef Živčák** [1]

1    Department of Biomedical Engineering and Measurement, Faculty of Mechanical Engineering, Technical University of Košice, Letná 9, 04200 Košice, Slovakia; mariana.trebunova@tuke.sk (M.T.); marek.schnitzer@tuke.sk (M.S.); lucia.bednarcikova@tuke.sk (L.B.); jozef.zivcak@tuke.sk (J.Ž.)

2    Department of Radiology, Faculty of Medicine, AGEL Hospital Košice-Šaca, Pavol Jozef Šafarik University in Košice, Lúčna 57, 04015 Košice-Šaca, Slovakia; tatiana.spakovska@nemocnicasaca.sk

*    Correspondence: darina.bacenkova@tuke.sk; Tel.: +42-1055-602-2358

**Abstract:** The global pandemic known as coronavirus disease 2019 (COVID-19) was caused by severe acute respiratory syndrome coronavirus 2 (SARS-CoV-2). This review article presents the taxonomy of SARS-CoV-2 coronaviruses, which have been classified as the seventh known human pathogenic coronavirus. The etiology of COVID-19 is also briefly discussed. Selected characteristics of SARS-CoV-2, SARS-CoV, and HCoV-NL63 are compared in the article. The angiotensin converting enzyme-2 (ACE-2) has been identified as the receptor for the SARS-CoV-2 viral entry. ACE2 is well-known as a counter-regulator of the renin-angiotensin system (RAAS) and plays a key role in the cardiovascular system. In the therapy of patients with COVID-19, there has been a concern about the use of RAAS inhibitors. As a result, it is hypothesized that ACE inhibitors do not directly affect ACE2 activity in clinical use. Coronaviruses are zoonotic RNA viruses. Identification of the primary causative agent of the SARS-CoV-2 is essential. Sequencing showed that the genome of the Bat CoVRaTG13 virus found in bats matches the genome of up to (96.2%) of SARS-CoV-2 virus. Sufficient knowledge of the molecular and biological mechanisms along with reliable information related to SARS-CoV-2 gives hope for a quick solution to epidemiological questions and therapeutic processes.

**Keywords:** SARS-CoV-2; SARS-CoV; HCoV-NL63; ACE2 receptor

## 1. Introduction

The first cases of an acute respiratory syndrome of unknown etiology were reported at the end of 2019 and beginning of 2020 in Wuhan, China. A new coronavirus was the causative agent of a very rapidly spreading outbreak of respiratory disease and potentially severe pneumonia. The genome of the virus causing the disease was isolated and sequenced. The new virus is genetically related to the severe acute respiratory syndrome coronavirus (SARS-CoV) virus, which caused severe to lethal pneumonia in 2003 in China and Canada [1,2]. The disease caused by the virus was officially named by the WHO as coronavirus disease 2019 (COVID-19). On 11 February, 2020, the International Committee on Taxonomy of Viruses (ICTV) announced the name of the new virus as severe acute respiratory syndrome coronavirus 2 (SARS-CoV-2). The new virus has been classified as the seventh known human pathogenic coronavirus [3,4].

### 1.1. Taxonomy of Human Coronaviruses

Pathogenic coronaviruses (CoVs) have originated in a variety of animal species, mainly bats, cattle, camels, civets, and mice [5]. In animals, they can cause serious respiratory, gastrointestinal, cardiovascular, and neurological diseases. Due to the high prevalence and wide distribution of CoVs as well as the high level of genetic diversity and frequent recombination of their genomes, these viruses pose a significant threat to human health [6]. Three

highly pathogenic coronaviruses of zoonotic origin, SARS-CoV, Middle Eastern respiratory syndrome coronavirus (MERS-CoV,) and SARS-CoV-2, which cause fatal respiratory diseases, have been reported in humans in the 21st century. The virus was taxonomically classified using information derived from a sequence based on family classification. According to the ICTV, SARS-CoV-2 is classified as follows: Coronaviruses belong the realm Riboviria, kingdom Orthornavirae, phylum Pisuviricota, class Pisoniviricetes, order Nidovirales, suborder Cornidovirineae, family Coronaviridae, subfamily Orthocoronavirinae, genus *Betacoronavirus*, subgenus *Sarbecovirus*, species *SARS-CoV-2* [7]. The current classification of coronaviruses contains many species in realm Riboviria, kingdom Orthornavirae, Phylum Pisuviricota, order Nidovirales, and suborder Cornidovirineae, which belong to the family Coronaviridae, and two Orthocoronavirinae subfamilies are divided into four genera: *Alphacoronavirus*, *Betacoronavirus*, *Gammacoronavirus*, and *Deltacoronavirus* [8]. In humans, disease is caused by seven known types of CoVs. Human coronavirus OC43 (HCoV-OC43), human coronavirus HKU1 (HCoV-HKU1), SARS-CoV-2, SARS-CoV, and MERS-CoV are *Betacoronaviruses*. Further genera that can cause disease in humans include *Alphacoronaviruses* human coronavirus NL63 (HCoV-NL63) and human coronavirus (HCoV-229) [9] (Figure 1).

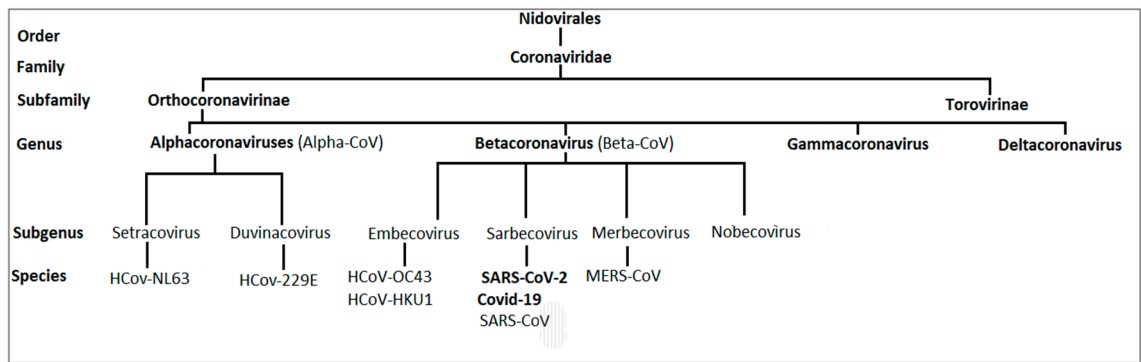

**Figure 1.** The classification of coronaviruses in realm Riboviria, kingdom Orthornavirae, phylum Pisuviricota, order Nidovirales, suborder Cornidovirineae, family Coronaviridae, and two subfamilies Orthocoronavirinae, is divided into four genera: *Alphacoronavirus*, *Betacoronavirus*, *Gammacoronavirus*, and *Deltacoronavirus*. Human coronavirus OC43 (HCoV-OC43), human coronavirus HKU1 (HCoV-HKU1), SARS-CoV-2, severe acute respiratory syndrome coronavirus (SARS-CoV), and Middle Eastern respiratory coronavirus (MERS-CoV) are *Betacoronaviruses*. human coronavirus NL63 (HCoV-NL63) and human coronavirus (HCoV-229) are *Alphacoronaviruses*.

### 1.2. Etiology of the Disease

The new contagious epidemic of atypical pneumonia, known as COVID-19, broke out in China in December 2019. The disease quickly spread from the city of Wuhan in the province of Hubei to other parts of China and soon thereafter worldwide. SARS-CoV-2 caused a global pandemic during 2020. The potential source of the infection was identified as a local animal and seafood market [10,11]. The clinical spectrum of disease symptoms varies, ranging from severe pneumonia to cases of mild upper respiratory tract illness. It was crucial to quickly obtain data on the clinical outcomes of critically ill patients with COVID-19 infection [12]. Common symptoms of the disease are dyspnea, hypoxemia, febrility, and a dry cough [3]. In severe cases, acute respiratory distress syndrome (ARDS), lymphopenia, and pro-inflammatory cytokine release syndrome may occur [4]. Elevated levels of serum interleukins IL-2, IL-6, IL-7, and IL-10, and tumor necrosis factor (TNF) have been reported [13]. The data suggest that immune system homeostasis plays an important role in the development of COVID-19 pneumonia. As reported by Chen et al., in tested COVID-19 patients, the leukocytes in the peripheral blood remain normal. Decreased lymphocyte [14] and platelet counts have been observed [15,16]. The number of granulocytes significantly decreases. Patients were found to have higher CD14 levels

than healthy people [17]. Following the occurrence of pathogenic SARS-CoV in 2003 and MERS-CoV in 2012, SARS-CoV-2 also causes severe pneumonia. It may even lead to an ARDS. In the human population, HCoV-229E, HCoV-OC43, HCoV-NL63, and HCoV-HKU1 act mostly on a local basis; they are responsible for less serious upper respiratory tract diseases. Depending on the season, they often occur in children. CoVs are detected in 4–6% of children hospitalized for acute respiratory infections and 8% of children examined on an outpatient basis. Children under the age of three years and children with associated cardiovascular disease are most commonly affected. Later reinfections are common, despite most children developing antibodies in childhood [18].

### 1.3. Morphology, Structure, and Replication of SARS-CoV-2

Coronaviruses are zoonotic RNA viruses [11,17]. The viral particles have a lipoprotein envelope. They are classed as enveloped viruses, which are characterized by the transmission of infection by air or droplets. CoVs contain single-stranded RNA with positive polarity, with the largest known RNA genome of 26–32 kilobases [19]. The genome size of SARS-CoV-2 varies from 29.8 kb to 29.9 kb [20]. The SARS-CoV-2 virion particle has a typical spherical shape with a diameter of 78 nm. The helical nucleocapsid is surrounded by a membrane. This is covered by surface spikes with a size of 20 nm, which gives the virion particles an appearance in an electron microscope resembling a solar corona [7].

#### 1.3.1. Nonstructural Proteins

SARS-CoV-2 encodes several unique group-specific open reading frames (ORFs) relative to other known coronaviruses. The replicase locus of SARS-CoV-2 contains two substantial ORFs, 1a and 1b, which are the largest part of the SARS-CoV-2 genome, spanning about 20 kb and two-thirds of the genome. It contains 16 non-structural proteins (nsps), termed nsp1 to nsp16. The replicase-locus-encoded nonstructural proteins (nsp1–nsp16) all have functions required for replication and transcription in the virus's life cycle. A smaller part of the genome encodes four structural proteins. Many nsps are involved in viral replication and represent potential targets for antiviral drugs. SARS-CoV-2 nsp12, the RNA-dependent RNA polymerase (RdRp), forms a large multiprotein complex with nsp7 and nsp8, is involved in RNA replication, and is part of a much larger replication and transcription complex (RTC). SARS-CoV-2 nsp10 exhibits close similarity with its SARS homologue. The ORF1a encodes a polyprotein (pp) 1a, whereas ORF1a and ORF1b together encode a C terminal-extended frameshift protein pp1ab. The polyproteins pp1a and pp1ab are first produced and then further processed via autoproteolysis by the viral-encoded polymerase in the virion into nsp1–nsp16 [21].

#### 1.3.2. Structural Proteins

The downstream ORFs of the RNA genome encode the four structural proteins: Spike (S), nucleocapsid (N), envelope (E), and membrane (M). These are essential for the assembly of the new RNA genome during replication. The S protein is the largest structural protein encoded by the RNA genome. The S protein functions in recognizing angiotensin converting enzyme 2 (ACE2) on the host cell. This protein consists of two functional units. The S protein may be a target for inhibition of viral entry and the development of antibody-based therapeutics to prevent the disease. The S protein has two functional units: S1 (bulb part) for receptor binding and S2 (stalk part) for membrane fusion; their functions may need other human receptors or proteases such as transmembrane protease, serine 2 (TMPRSS2), and furin to activate and facilitate viral entry into host cells. The E protein is a small membrane protein that plays a role in virion assembly and has additional effects on the infected cells of the host. The E protein alters coronavirus virulence by regulating the cell stress response and apoptosis during host cell infection. The M protein is the most abundant in the viral envelope. It provides structural support for the virion and plays a role in the assembly and construction of virion particles. N complexes with the genome

RNA to form the virial nucleocapsid within the viral membrane and interact with the M protein in the assembly. The N protein may play a role in replicating RNA [22] (Figure 2).

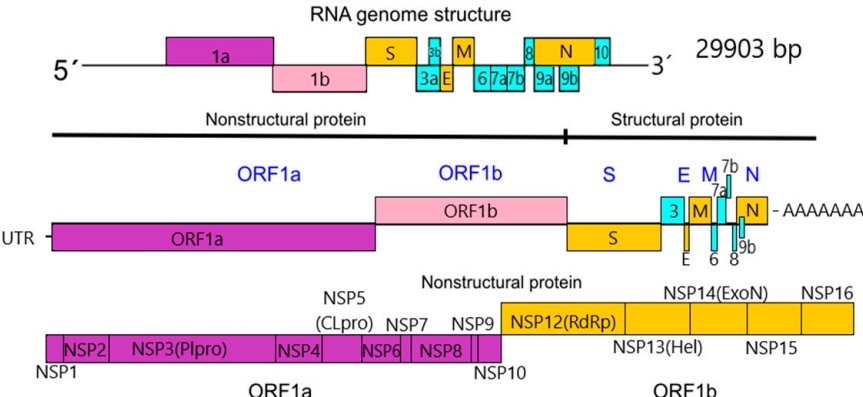

**Figure 2.** Schematic RNA genome structure SARS-CoV-2. The replicase locus of SARS-CoV-2 contains two substantial open reading frames (ORFs) and four structural proteins. The ORF1a and ORF1b locus encodes 16 nonstructural proteins. Structural proteins, the spike (S) protein is the largest structural protein encoded by the RNA genome. The envelope (E) protein is a small membrane protein that is anchored on the viral envelope and plays a role in the virion assembly and has additional effects on the infected cells of the host. The membrane (M) protein is a multi-membrane spanning protein and the most abundant in the viral envelope. It provides structural support for the virion and plays a role in the assembly and construction of virion particles. Nucleocapside (N) protein complexes with the genome RNA to form the virial nucleocapsid within the viral membrane and interact with the M in the assembly. The N protein may play a role in replicating RNA.

### 1.3.3. Replication

The SARS-CoV-2 virus primarily attacks respiratory epithelial cells, alveolar pneumocytes, mainly type II [23], as well as intestinal tract epithelial cells. Kidney and heart tissue is secondary [24,25]. CoVs enter cells by specifically binding to the cell receptor of the invaded cell and subsequent fusion with the membranes. *ACE2* was specified as the binding receptor between the host cell and the spike protein through the receptor-binding domain (RBD) in SARS-CoV-2. The susceptibility of the host is dependent on the affinity of the virus to its receptor. SARS-CoV-2 recognizes ACE2 more efficiently than SARS-CoV because of its 10- to 20-fold increased binding affinity to ACE2. This binding leads to the entry of the virus into the host cell and to the activation of the S protein by a specific TMPRSS2, of the host cell. The S protein, which consists of two subunits and S1, catalyzes the fusion of the S2 subunit. Upon binding of the virion to the target cell, the protease of the cleaved TMPRSS2 protein opens the top protein of the virus and reveals the fusion peptide in the S2 subunit and the ACE2 host receptor. The virus enters the cytoplasm of the cells and fuses with the membrane of the host cell. Following the fusion, an endosome is formed around the virion. Afterward, RNA is released into the cell [8,26]. SARS-CoV-2 viruses are positive-sense, single-stranded RNA viruses, which employ a complicated pattern of virus genome length RNA replication as well as transcription of genome length and leader-containing subgenomic RNAs. Replication of RNA viruses occurs in the cytoplasm of the host cell in several consecutive phases. Subsequently, the virus genome is exposed by the disintegration of the viral protein envelope and the transcription of the viral RNA into the RNA of the host cell. The first part of the genome to be transcribed is a viral RNA polymerase, called RNA replicase, which transcribes copies of the full-length genome in negative strands. These are used as templates to produce *messenger RNA* (mRNA) that transcribe viral genes. Transcripts are produced by viral RNA polymerase activity. The viral RNA polymerase interacts with a transcriptional regulation sequence (TRS) located between the viral genes, which enables a link between the 5′ leader sequence and the beginning of each gene. The mechanism of replication has not yet

been fully described. The genome from which the four major structural proteins arise, as well as a number of non-structural proteins (nsps) from two alternative reading frames is ORF1$\alpha$, which generates polyprotein 1a followed by 11 nsps and ORF1b, resulting in polyprotein 1ab, which is further processed to 16 nsps. [27]. The S protein is 150 kDa and N-glycosylated. The S protein trimer forms a distinct structure on the surface of the virus [28]. The trimeric S protein is cleaved by a host furin-like protease into two domains: S1 and S2. S1 provides the binding site to the receptor, whereas S2 provides structural aid in the form of support for a functional S protein [24]. The M protein has a size of 25 to 30 kDa. It consists of three transmembrane domains. The M protein has an N-terminal ectodomain and a C-terminal end domain. It gives the virion an overall curved shape. The CoV envelope (E) protein is a small, transmembrane E protein and it is 8–12 kD in size and is thought to function in SARS-CoV-2 through involvement in several aspects of the virus' life cycle, such as assembly, budding, envelope formation, and pathogenesis. It contains an N-terminal ectodomain and a C-terminal endodomain. The E protein is thought to be a pentameric structure when the protein is contained in micelles. Here, the C-terminal end of the protein forms an extramembranous alpha-helix. The N protein is part of the nucleocapsid; it contains an N-terminal domain and the C-domain has the ability to bind to RNA. The N protein is phosphorylated, which increases its affinity for RNA; together, they form a complex structure [27]. The N protein aids in the re-creation of the viral envelope by the M protein. It also mediates the binding of RNA replicase and reverse transcriptase [29]. In cells infected with SARS-CoV-2, granular regions containing viral RNA and proteins have been observed. These areas have not been observed in other CoVs strains. They are thought to be viral translation centers. Translated structural proteins translocate into endoplasmic reticulum (ER) membranes and transit through the ER-to-Golgi intermediate compartment, where interaction with N-encapsidated, newly produced genomic RNA results in budding into the lumen of secretory vesicular compartments. Virions are secreted from the infected cell by exocytosis. [30,31] (Figure 3).

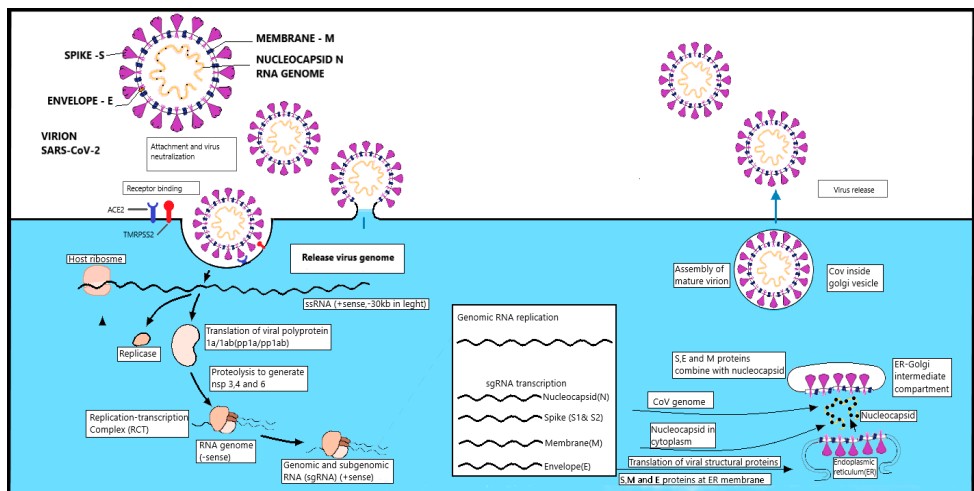

**Figure 3.** The coronavirus virion and life cycle. The SARS-CoV-2 virion consists of structural proteins, spike—S, envelope—E, membrane—M, nucleocapsid—N, and specific binding of the S protein to ACE2 receptor and the cofactor cell surface serine protease TMPRSS2. Following entry, the release genomic RNA subject it to the immediate translation of two large open reading frames, ORF1a and ORF1b. The resulting polyproteins pp1a and pp1ab are co-translationally and post-translationally processed into the individual non-structural proteins (nsps) that form the viral replication and transcription complex. For viral genomic RNA replication and transcription of subgenomic mRNAs (sg mRNAs) comprise the characteristic set of coronavirus mRNAs. Translated structural proteins translocate into endoplasmic reticulum (ER) membranes and transit through the ER-to-Golgi intermediate compartment, where interaction with N-encapsidated, newly produced genomic RNA results in budding into the lumen of secretory vesicular compartments. Virions are secreted from the infected cell by exocytosis.

### 1.3.4. Spike Protein and the Function of TMPRSS2

Significant similarity exists between SARS-CoV and SARS-CoV-2. The level of similarity in the genome sequence has been confirmed to be 79.5% [13]. The E protein has been found to be present in both. The E protein has been found to be essential for the ability of other coronaviruses to cause disease. It is encoded by the subgenomic RNA (sgRNA), which is among the transcripts with the highest copy number. This may be due to structural differences in the spike of the S protein. Recognition of the host cell occurs through the S protein and its receptor by a virion. The SARS-CoV-2-binding domain is at the S1 domain at its N-terminus [8]. For SARS-CoV, it is also on the S1 domain, but at its C-terminal end [32]. The interaction between the S protein and its receptor is responsible for the species specificity and the tissue tropism of the virus. Many CoVs use peptidases as cellular receptors. ACE2 is specific for SARS-CoV and HCoV-NL63. The surface of RBD S1 uses 14 amino acids that recognize ACE2. Of the 14, 8 are strictly bound to SARS-CoV-2. This is different from MERS-CoV, which is recognized by the dipeptidyl peptidase 4 (DPP4) of the host cells [33]. Upon receptor binding, the proteolytic cleavage of coronavirus S proteins by host-cell-derived proteases is essential to permit fusion. The virus enters the cytosol of the host cell via an acid-dependent proteolytic cleavage of the S protein by cathepsin, a specific TMPRRS2, followed by the fusion of viral and cell membranes. SARS-CoV-2 has been shown to use the cell-surface serine protease TMPRSS2 for priming and entry, although endosomal cysteine proteases cathepsin B (CatB) and CatL can also assist in this process. S protein cleavage occurs at two sites within the S2 portion of the protein, with the first cleavage for separating the RBD and fusion domains of the S protein. [27]. The second cleavage results in binding of the fusion peptide. Subsequently, the plasma membrane overflows. The exposed S2 fusion peptide aids in the fusion of membranes and the subsequent endocytosis and release of the viral genome into the host cell [34]. In addition to ACE2 entry factors, such as cellular glycans and integrins, may impact the observed phenotypic differences of SARS-CoV and SARS-CoV-2. Viruses infect host cells by initially binding to the surfaces of the cells. There are many pathogens, including viruses and bacteria that have the ability to use integrins with different mechanisms for invading cells. As multifunctional heterodimeric cell-surface receptor molecules, integrins have been shown to usefully serve as entry receptors for a plethora of viruses. The most common of these motifs is the peptide sequence for binding integrins, RGD (Arg-Gly-Asp). Virus-integrin binding is shown to facilitate adhesion, cytoskeleton rearrangement, integrin activation [35]. Virus binding only ensures viral proximity to cells. The SARS-CoV-2 spike glycoprotein RGD lies in the receptor-binding domain (amino acids 319 to 541) [36]. In order to bind integrin, the RGD motif must be present at the surface of the protein. ACE2 binding induces conformational changes involving one of the receptor-binding domains of the trimer. After a confirmatory change, the ACE2 binding domain and the RGD region are exposed. This process is the step that kick-starts the whole cascade of events, resulting in the eventual internalization of the virus. A cellular process reminiscent of viral infection revealed an endocytosis mechanism by which the cell surface heparan sulfate (HS) facilitates receptor-mediated uptake of protein assemblies bearing excess positive charges Heparan sulfate facilitates spike-dependent viral entry [37]. These mechanisms of virus penetration into the host cell are similar in SARS-CoV-2, SARS-CoV, and HCoV-NL63. A detailed description of the molecular events in the penetration of a virus or cell may be helpful in the development of specific therapeutic approaches [26].

### 1.3.5. RAAS Complex

Severe COVID-19 infection may lead to multi-system organ dysfunction and failure. Multiple case studies demonstrated that underlying hypertension and diabetes mellitus are more common in patients with severe COVID-19 infection than in patients with mild infection. Hypertension has been described as a major risk factor for severe COVID-19 disease. Medications modulating the renin-angiotensin system (RAAS) are used to treat hypertension, therefore concerns about the use of these medications in patients with

COVID-19 infection are reasonable [38]. RAAS inhibitors, angiotensin-converting enzyme (ACE) inhibitors and angiotensin receptor blockers (ARBs) are commonly prescribed anti-hypertensive drugs. The RAAS is an enzymatic cascade that regulates key physiological processes in the human body and plays a critical role in maintaining blood pressure homeostasis, as well as fluid and salt balance. Angiotensin is a peptide hormone that causes vasoconstriction and blood pressure increase. Angiotensin is cleaved at the N-terminus by renin to result in angiotensin I, which will later be modified to become angiotensin II. ACE2 is integral membrane-bound monocarboxypeptidase protein that serves as a counter regulatory role in cardiovascular homeostasis. ACE2 converts Angiotensine II to Angiotensine (1–7) thereby creating a cytoprotective environment through anti-inflammatory, anti-fibrotic, anti-proliferative, and vasodilatory effects via MAS receptor. The classical RAAS axis, formed by ACE, angiotensin II, and angiotensin receptor type 1 (AT1), activates several cell functions and molecular signalling pathways related to tissue injury and inflammation. SARS-CoV-2 enters lung cells via the ACE2 receptor [38,39]. Therapy in patients with COVID-19 has raised concerns about the use of RAAS inhibitors, which may affect the occurrence of ACE2 on the cell membrane. It may also impact increased ACE2 expression following therapy in blocking ACE inhibitors [26]. Physicians have questioned whether this may affect the virulence of SARS-CoV-2. The use of angiotensin receptor blockers (ARBs) losartan, candesartan, and valsartan is common worldwide. Patient data are too limited to support or refute these assumptions. However, the effects of RAAS blockers on ACE2 levels suggest that the omission of ACE2 blockers may be detrimental rather than beneficial in patients with lung injury [40]. The ACE2 reduces angiotensin II effects on vasoconstriction, sodium retention, and fibrosis. [41]. Circulating levels of soluble ACE2 are low and ACE2 function in the lung is relatively minimal under normal conditions, however may be increased in some clinical cases. The effect of RAAS inhibitors on ACE2 in humans is unknown due to the fact that ACE inhibitors and ARBs have different reactions on angiotensin II, the primary substrate of ACE2 [34]. The effects of these substances on ACE2 levels and their activity may therefore vary. Despite considerable structural homology between ACE and ACE2, their active enzyme sites are different. As a result, ACE inhibitors do not directly affect ACE2 activity in clinical practice. Experimental animal models have shown mixed findings related to the results of ACE inhibitors on ACE2 levels or tissue activity. Similarly, animal models have produced inconsistent findings related to outcomes of ARBs on ACE2, with some indicating that ARB may increase messenger RNA expression. In post-mortem autopsy, heart tissues from several patients who succumbed to SARS-CoV and heart samples had detectable viral SARS-CoV genome. These patients were also characterized by reduced myocardial ACE2 expression, which further confirmed that ACE2 expression was decrease after virus infection. From the recent information, it has been found that RNA sequencing has revealed an abundance of ACE2 expression in the ileum, heart, kidney, bladder, airways, lung, esophagus, stomach, and liver. Current studies have shown that both ACE inhibitors and ARB affect the mRNA expression of ACE2. For example, ACE inhibitors, lisinopril and enalapril, can increase ACE2 mRNA expression in heart. ARB drugs, such as losartan and olmesartan, increase myocardial ACE2 mRNA by approximately 3-fold. The change of protein levels is not always consistent with the mRNA levels, sometimes even in the opposite direction. Many factors lead to the inconsistencies between mRNA and protein expression levels, including microRNA regulation, translation, post-translational modification, protein transport, and degradation [42] (Figure 4).

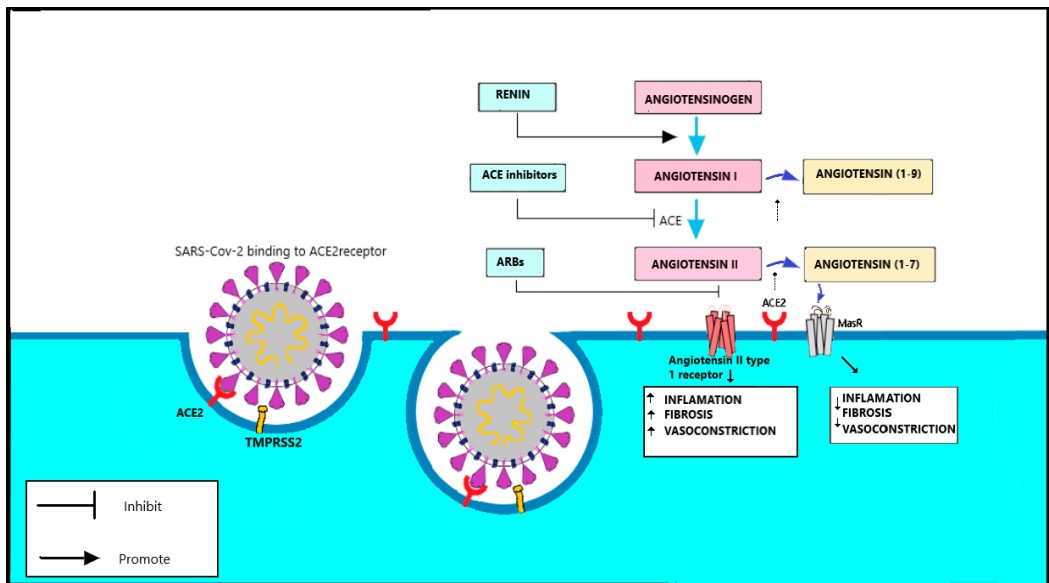

**Figure 4.** Renin–angiotensin–aldosteron system (RAAS). SARS-CoV-2 mainly enters human cells via the receptor angiotensin converting enzyme 2 and priming S protein in TMPRSS2 protease. Angiotensin-converting enzyme (ACE2) is expressed on major SARS-CoV-2 target cells, the pulmonary alveolar epithelial cells. ACE2 is an enzyme involved in the RASS, whose role is to regulate and counter angiotensin-converting enzyme (ACE), reducing the amount of angiotensin II and increasing angiotensin (1–7), making it a promising drug target for treating cardiovascular diseases. The classical RASS axis, formed by ACE, angiotensin II, and angiotensin receptor type 1 (AT1), activates several cell functions and molecular signaling pathways related to tissue injury and inflammation. The RAAS axis composed of ACE2, Angiotensin (1–7), and Mas receptor (MasR) exerts the opposite effect concerning the inflammatory response and tissue fibrosis.

### 1.4. Animal Origin of SARS-CoV

Coronaviruses, occurring in different animal species, have widely diversified host types. In many species of livestock, they cause diseases of the gastrointestinal tract and respiratory tract. Several studies have found the presence of CoVs RNA in animal fecal samples [43] and virus replication in the intestinal tract [44]. A genetic match exists between bat CoVs and human CoVs. Bat CoVs are most often detected in the feces. This suggests an initial attack on the intestinal tract in animals. In humans, they are predominantly respiratory tract pathogens [45]. CoV infections mainly cause respiratory diseases in humans. The ubiquity and wide diversity of CoVs associated with bats have led to the assumption that bats are the primary hosts of coronaviruses [46].

#### 1.4.1. SARS-CoV-2 and Ist Natural Hosts

Sequencing showed that the genome sequences of the bat CoV RaTG13 virus of *Rhinolophus affinis* bats in the family *Rhinolophidae* matches SARS-like-CoV by up to 96.2% and it is closest to SARS-CoV-2 [10,19]. In case of epidemics due to zoonoses in human, in addition to the original primary host, the presence of another intermediate host is presumed. If the virus is present in the intermediate host for a long time, it can become its natural host. A frequent secondary host is smaller carnivores, which may be present in nature but also as partially domesticated animals. The original form of bat CoV RaTG13 in *Rhinolophus affinis* was able to increase ACE2 receptor binding affinity in other animals after mutation of the RBD. For SARS-CoV-2, there are indications that the bat CoV has undergone a recombination event with CoV from pangolins, where it acquired the RBM domain, which is crucial for binding to the human ACE2 receptor [34]. CoVs have been detected in ferrets and pangolins (*Manis pentadactyla)*. In contrast, the affinity for ACE2 was reduced in rodents [47]. It has been hypothesized that the secondary carrier is a rare species of Chinese pangolin, *Manis pentadactyla* [48,49], which is hunted in China for meat and medical uses. Despite the genetic agreement, doubts exist about the direct transmission

of the coronavirus from pangolins to humans. Recently, genome differences in the RRAR sequence have been demonstrated in the area of the S protein (the S1–S2 boundary), which is responsible for ACE2 recognition, related to SARS-CoV-2. [50].

### 1.4.2. Severe Acute Respiratory Syndrome SARS-CoV and Its Natural Hosts

A highly infectious and fatal coronavirus disease was reported in 2003. The virus crossed the species barrier of the zoonotic reservoir to infect the human population [51]. It is assumed that the main source of the pathogen was found in local markets of Guangdong province [52], China. Cases of severe atypical pneumonia with subsequent respiratory failure were observed in Guangdong. The disease spread rapidly mainly due to a lack of population immunity to the new pathogen [53]. Human-to-human transmission occurred in close contact. Medical and nursing staff were mainly at risk and the virus spread further among family members. Transmission also occurred indirectly through infected respiratory secretions [23]. The virus was identified by researchers in March 2003 [54]; it was isolated on tissue cultures by electron microscopy and PCR sequencing of the virus genome [30]. The isolated virus was named SARS-CoV. The WHO defined clinical symptoms for the detection of suspected cases: Temperatures above 38 °C, a confirmed positive X-ray finding of pneumonia, RNA detection of the presence of the virus, and confirmed antibodies against SARS-CoV virus. The incubation period of the disease was established as 6–10 days. Lymphopenia and occasionally thrombocytopenia were reported based on hematological parameters. Elevated lactate dehydrogenase and creatine kinase levels were observed from biochemical values, often associated with a worse prognosis [55]. Zoonotic transmission was confirmed as the source of the disease. The primary natural reservoir is most likely the bat genus *Rhinolophus*. *Rhinolophus sinicus* was analyzed in the Chinese provinces of Hubei and Guangdong; coronaviruses genetically similar to SARS-CoV were detected in 22 out of 59 individuals. In addition, serum from *Rhinolophus sinicus* contained antibodies specific to the human SARS-CoV N protein, antibodies identical to the SARS-CoV N protein, were also present. Genome sequencing showed that the genome of bat CoVs is genetically nearly identical to SARS-CoV isolated in humans and local civet species. Therefore, the most likely secondary vector was the palm civet. About 80% of civets (*Paguma larvata*) from animal markets in *Guangzhou* contained significant levels of antibodies to SARS-like-CoV [56]. The isolated coronavirus is genetically identical to SARS-CoV by up to 99%. In rare cases, genetic relatedness has been demonstrated to coronaviruses in the racoon dog (*Nyctereutes procyonoides*). Antibodies were detected in the Chinese ferret-badger *Melogale moschata*. The sequences were similar at 88–92%. Most of the variable regions were located at the 5′ end of the S gene, which encodes the S1 domain responsible for the receptor binding region. Many SARS-CoV samples from humans isolated during the later stages of the epidemic in 2002–2003 had a 29 nt deletion in this region; this deletion is absent in civet isolates or human isolates from the early phase of foci. Bat viruses do not contain a deletion of 29, suggesting that SARS-CoV and animal SARS-like-CoV share a common ancestor. The SARS-CoV uses the ACE2 receptor to enter the cell [57]. The ability to effectively use the ACE2 receptor in humans and civets appears to be a major difference from other zoonotic CoVs. This is also the difference between SARS-CoV in humans and SARS-like-CoV virus in bats. This explains the limitation of the spread of the virus between individual species [58].

### 1.4.3. HCoV-NL63 and Its Natural Hosts

The human coronavirus HCoV-NL63 was identified in the Netherlands in late 2004 in a seven-month-old child with bronchiolitis and fever. HCoV-NL63 is a member of the group of alphacoronaviruses and is most closely related to HCoV-229E. The differences between HCoV-NL63 and HCoV-229E are prominent. First, they share, on average, only 65% sequence identity. Before the discovery of HCoV-NL63, it was generally thought that all group I coronaviruses use CD13 (aminopeptidase N) as the receptor, as was described for HCoV-229E. NL63 is not able to use CD13 as a receptor for cell entry, but unusually uses the

ACE2 receptor. The virus is capable of replication in monkey kidney cells, whereas HCoV-229E is not. SARS-CoV, being a betacoronavirus, is also able to replicate in monkey kidney cells. This virus was subsequently identified in various countries, indicating a worldwide distribution. The coronavirus was isolated from a patient sample in the supernatant of the renal cell line tMK with a cytopathic effect on the LLC-MK2 cell line (monkey kidney cell line) using the Virus Discovery—complementary DNA—amplified fragment length polymorphism (cDNA-AFLP) (VIDISCA) method [59]. To determine the etiology of HCoV-NL63, prevalence was tested in a group of 139 subjects and positivity was detected in four patients, which is approximately 3% [60]. Viral infection has been confirmed worldwide. It is associated with many common symptoms and diseases. Manifestations of the disease include mild-to-moderate upper respiratory tract infections, severe lower respiratory tract infections, and bronchitis. The virus occurs mainly in young children, the elderly, and immunocompromised patients. It also has a seasonal occurrence in Europe. A study in Amsterdam estimated the presence of HCoV-NL63 as approximately 4.7% of common respiratory diseases [61]. It is currently assumed that the virus was modified and transmitted primarily from infected bats and subsequently from palm civets to humans. The virus could have been transmitted from bats to civets that were in contact in animal markets in China [9]. The first evidence of transmission of HCoV-NL63 virus was found in the fecal samples of European and African bats belonging to the Vespertilionidae family [62]. Isolated CoVs are genetically related to HCoV-NL63. Viruses with some degree of similarity to HCoV-NL63 have been found in the tricolored bat *Perimyotis subflavus*, namely bat CoV ARCoV.2 [63]. This study provided sufficient evidence that HCoV-NL63 replicates on a lung cell line from a tricolored bat, as confirmed by RNA detection and nucleocapsid protein production. Results were published by Tao et al. [64], who described a high genetic similarity of coronaviruses of Kenyan bat species in the family Hipposideridea, with up to 90% similarity of the genomic sequences between BtKYNL-63-9a and HCov-NL63. They also described a possible recombination between BtKYNL-63-9a and HCoV-229E. Both of these HCoVs are alphacoronaviruses. Comparison of the properties of human CoVs shows common characteristics (Table 1).

**Table 1.** Comparison of human coronaviruses and natural animal hosts and cellular receptors for viral entry.

| Type of Coronavirus | Cell Surface Receptor | Primary Animals Hosts SARS-Like-CoV | Secondary Animals Hosts SARS-Like-CoV | Human Discovery |
|---|---|---|---|---|
| SARS-CoV-2 (Covid-19) | Angiotensin converting enzyme 2 (ACE2) | Bats genus *Rhinolophus*, *Rhinolophus affinis* [19] | Chinese pangolins, *Manis petadactyla* [48,49] | 2019 |
| MERS-CoV | Dipeptidyl peptidase 4 (DPP4) | Bats *Pipistrellus abramus* *Pipistrellus pipistrellus* [61,62] | Camels, *Camelus dromedarius* [58,61] | 2012 |
| HCoV-NL63 | Angiotensin converting enzyme 2 (ACE2) | Bats *Permyotis subfavus* [58,63,64] | Palm civets, *Paguma larvata* [9] | 2004 |
| SARS-CoV | Angiotensin converting enzyme 2 (ACE2) | Bats genus *Rhinolophus*, *Rhinolophus sinicus* [5,51,63] | Palm civets, *Paguma larvata* [51,52,56] | 2002 |
| HCoV-229E | Aminopeptidase N (APN) | Bats, order Chiroptera [58,64] | Camelids, *Camelidae* [61] | 2009 |
| CoV-HKU1 | Dipeptidyl peptidase 4 (DPP4) | Rodents, order Rodentia [61] | ? | 2004 |
| HCoV-OC43 | N-Acetylneuraminic acid (Neu5Ac or NANA) | Rodents, order Rodentia [61] | Bovines, subfamily *Bovinae* [58,61] | 2002 |

## 2. Conclusions

Prior to 2003, only mild forms of coronavirus diseases were reported in humans. They caused inflammation of the upper and, rarely, lower respiratory tract. CoVs were much more common as pathogens of wild animals [15]. After 2003, severe pandemic forms of human CoVs emerged: SARS-CoV, MERS-CoV, and SARS-CoV-2 [65]. Zoonotic transmission was identified in all human CoVs. The primary hosts of CoVs are several genera of bats, transmissible to humans via an intermediate host [63,66]. The issue of SARS-CoV-2 transmission remains open. The disease is primarily transmitted from bat CoV RaTG13 in Chinese bats of the genus *Rhinolophidae* with a significant match in gene nucleotides, via a probable secondary host, the rare pangolin species, which is has not been definitively confirmed due to differences in the genome in the S region of the binding spike protein [19]. Illnesses from COVID-19, SARS-CoV, and the less pathogenic HCoV-NL63 have several characteristics in common. However, the mortality rate for SARS-CoV is much higher than for SARS-CoV-2. [67]. They cause respiratory diseases with a complicated and atypical course of pneumonia. HCoV-NL63 is an endemic coronavirus strain [60]. The reduced pathogenicity of HCoV-NL63 suggests that ACE2 binding by the virus is not the only factor that determines the severity of viral pathogenicity. The biggest difference between the viruses is that SARS-CoV has caused local outbreaks of severe pneumonia, whereas SARS-CoV-2 has affected the world. These viruses infect alveolar epithelial cells by occupying the ACE2 receptor on the cell surface. In the RBD of the S protein, the amino acid similarity between SARS-CoV-2 and SARS-CoV is only 73% [68,69]. The virion to the invaded cell occurs through the recognition site of the S protein receptor. The RBD of SARS-CoV-2 is at domain S1 at its N-terminal end; that of SARS is also in the S1 domain, but at its C-terminal end. Differences in ACE2 binding may result in more efficient occupation of the host cell receptor. The main difference between SARS-CoV-2 and SARS-CoV is the higher infectivity of the new virus. Previous studies have shown that the RBD of the SARS-CoV-2-S1 subunit has approximately 10- to 20-fold higher affinity to ACE2 than SARS-CoV RBD. SARS-CoV-2 S also has a potential furin cleavage site in the S1/S2 region, which is unique to SARS-like CoV. The CoV infects the target cell by either cytoplasmic or endosomal membrane fusion. Therefore, the fusion capacity of CoV-S is a leading indicator of the infectivity of the corresponding virus [70]. The role of proteases in the penetration of SARS-CoV-2 is essential for activation and recognition to the cell surface and, secondarily, for the penetration of the virus into the cell. The action of the membrane protease TMPRSS2 plays an important role, with fusion of the cell surface membrane and the virus subsequently penetrating the cytoplasm of the host cell.

Of interest in therapy for hypertensive patients with COVID-19 is the question of continuing therapy with RAAS inhibitors or omitting ARB drugs. Serological levels of ACE2 are usually low. At present, the effects of RAAS inhibitor (ARB) therapy on ACE2 in humans are currently unknown. ACE inhibitors and ARBs have different effects on angiotensin II, the primary substrate of ACE2. The effects of these substances on ACE2 levels and activity may therefore vary. Despite the considerable structural homology between ACE and ACE2, their enzyme sites are different; therefore, ACE inhibitors do not directly affect ACE2 activity. The use of ACE inhibitors in ARB-type drugs is common worldwide. The effect of RAAS blockers on ACE2 levels suggests that omission of ACE blockers may be detrimental rather than beneficial in patients with lung injury. At this time, there are no data supporting RAAS antagonists worsen the prognosis of patients with COVID-19. Other studies have shown evidence that there is no association between RAAS antagonists and COVID-19. The joint statement from the American College of Cardiology and other professional societies strongly recommends that physicians should not initiate or withdraw RAAS-related treatments to COVID-19 infected patients with cardiovascular disease. [71]. Knowing the molecular and biological mechanisms and particularly the connection of information available regarding SARS-CoV-2 gives hope for a quick solution to epidemiological questions and therapeutic processes [26].

**Author Contributions:** Conceptualization, supervision, funding acquisition, J.Ž., project administration, M.T.; software, visualization T.Š., M.S., and L.B, formal analysis and investigation writing—original draft preparation D.B., M.S., and L.B. All authors have read and agreed to the published version of the manuscript.

**Funding:** This research was funded by the Educational Grant Agency of the Ministry of Education, Science, Research, and Sport of the Slovak Republic, grant number KEGA 023TUKE-4/2020, KEGA 041TUKE-4/2019; Slovak Research and Development, Agency grant number PP-COVID-20-0025; and CEMBAM—Centrum medicínskeho bioaditívneho výskumu a výroby, grant number ITMS2014+: 313011V358.

**Institutional Review Board Statement:** Not applicable.

**Informed Consent Statement:** Not applicable.

**Data Availability Statement:** Not applicable.

**Acknowledgments:** This work was supported by the Slovak Research and Development Agency under contract no. PP-COVID-20-0025.

**Conflicts of Interest:** The authors declare no conflict of interest.

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
