# Peer review of "Comparison of Selected Characteristics of SARS-CoV-2, SARS-CoV, and HCoV-NL63"

_applsci, doi:10.3390/app11041497_

Round 1
Reviewer 1 Report
Reviewer #1:
Coronavirus disease 2019 (COVID-19) has emerged as a new world pandemic, infecting millions of people with a substantial mortality. There is significant interest in understanding several characteristics of SARS-CoV-2 and comparison with SARS-CoV and HCoV-NL63.
Recently publications show several vaccine candidates in Clinical trials and this review could be help in understand this pandemic.
In this manuscript, by Bacenkova et al titled "Comparison of selected characteristics of SARS-CoV-2, SARS-CoV and HCoV-NL63". The authors performed an analysis of the aethiology of the disease, morphology, structure and replication of SARS-CoV-2, Spike protein and the function of TMPRSS2, RAAS complex, animal origin of SARS-like-CoV.
There are several concerns that to be addressed.
This manuscript is well written and sites key findings in the field, therefore it will be helpful for investigators entering into coronavirus/COVID-19 research. The study would benefit the section on general aspects concern to COVID-19 disease. Comments to improve the clarity of the manuscript are provided below.
Comments for the authors' consideration:
1.
- Double check small typing errors in all manuscript. For example: Line 64, line 108 in figure 1, SARS-Cov change to SARS-CoV.
- To use correctly SARS-CoV-2 and COVID-19. For example, in the line 86. I think is more appropriate mentioned SARS-CoV-2 versus COVID-19.
- Space in line 128 between 5’leader.
- In line 210 Figure 1 change to figure 3.
- Line 272. Ther change to The.
- Line 293. Cov change to CoV. Also in line, 332.
- Line 328. NL6 change to NL63.
- Please add a section explaining the evolution of structural and Nonstructural proteins in these viruses and your global distribution).

Author Response
Dear Reviewer , 01/04/2021
Thank you for giving us the opportunity to improve and resubmit our manuscript:
Comparison of selected characteristics of SARS-CoV-2, SARS-CoV and HCoV-NL63, Darina Bačenková, Marianna Trebuňová, Tatiana Špakovská and Jozef Živčák.
Please find enclosed the revised manuscript.
The manuscript has been revised according to the comments raised by the reviewer to the best of our ability. Changes to the manuscript are underscored through "Track Changes" function in
Microsoft Word. The manuscript was checked by an English editor from the editorial office.
We would like to thank the reviewer for the constructive and competent criticism, and we hope that
our manuscript will be acceptable for publication.
Sincerely, Darina Bačenková
Comments to the Author and answer.
- In line 64, line 108 in figure 1, SARS-Cov change to SARS-CoV.
Answer: I fixed it.
- In line 86 correctly SARS-CoV-2 and COVID-19. Is more appropriate mentioned SARS-CoV-2 versus COVID-19.
Answer: I fixed it.
- In line 128 Space between 5’leader.
Answer: I fixed it.
- In line 210 Figure 1 change to figure 3.
Answer: I fixed it. Figure 1 changed to Figure 4
- In line 272. Ther change to The.
Answer: Corrected.
- In line 293. Cov change to CoV. Also in line, 332.
Answer: I fixed it.
- In line 328. NL6 change to NL63.
Answer: Corrected.
- Please add a section explaining the evolution of structural and nostructural proteins in these viruses and your global distribution.
Answer: We have added a section on structural and non-structural proteins with literature and pictures to the text.
Reviewer 2 Report
In this review, Bacenkova and colleagues aim to compare SARS-CoV-2, SARS-CoV and HCoV-NL63. Unfortunately, the text is extremely different to read. Text is awkwardly phrased. A thorough revision is rquired
- Not clear this a review from reading the abstract.
- The manuscript needs thorough revisions (grammar and spellings mistakes were found). Stylistically, the review is difficult to read, with poor flow. The sentence structure makes interpretations of the text difficult for this this reader.
- Line 174, the authors mention integrins may play a role in entry. Please elaborate and provide references.
- Spike protein and the function of TMPRSS2 is used as a subtitle twice (1.4)
- This sentence "ACE2 is specific for SARS-CoV and HCoV-NL63" should be revised to include SARS-CoV-2
- Poorly organized.
- The review fails to systematically compare the three viruses it mentions in the title. There is limited discussion of HCoV-NL63.
- I would recommend submitting the manuscript to a professional editing service before resubmitting.
Author Response
Dear Reviewer , 01/04/2021
Thank you for giving us the opportunity to improve and resubmit our manuscript:
Comparison of selected characteristics of SARS-CoV-2, SARS-CoV and HCoV-NL63, Darina Bačenková, Marianna Trebuňová, Tatiana Špakovská and Jozef Živčák.
Please find enclosed the revised manuscript.
The manuscript has been revised according to the comments raised by the reviewer to the best of our ability. Changes to the manuscript are underscored through "Track Changes" function in
Microsoft Word.
We would like to thank the reviewer for the constructive and competent criticism, and we hope that
our manuscript will be acceptable for publication.
Sincerely, Darina Bačenková
Response to the reviewers Reviewer: 2
Comments to the Author and answer.
- Not clear this a review from reading the abstract.
Answer: The abstract has been reviewed and revised.
- The manuscript needs thorough revisions (grammar and spellings mistakes were found). Stylistically, the review is difficult to read, with poor flow. The sentence structure makes interpretations of the text difficult for this this reader.
Answer: The article has been edited in English. We have supplemented the article with a proposed section on structural proteins and non-structural proteins.
- Line 174, the authors mention integrins may play a role in entry. Please elaborate and provide references.
Answer: We supplemented the article with a proposed section on integrins with literature.
- Spike protein and the function of TMPRSS2 is used as a subtitle twice (1.4)
Answer: Corrected.
- This sentence "ACE2 is specific for SARS-CoV and HCoV-NL63" should be revised to include SARS-CoV-2.
Answer: Corrected.
- Poorly organized.
Answer: We have adjusted the organization of the article according to the recommendations.
- The review fails to systematically compare the three viruses it mentions in the title. There is limited discussion of HCoV-NL63.
Answer: We have supplemented the article with a proposed section on HCoV-NL63.
I would recommend submitting the manuscript to a professional editing service before resubmitting.
Answer: The manuscript was checked by an English editor from the editorial office. We have supplemented the article with a proposed section on structural proteins and non-structural proteins.
Reviewer 3 Report
I read with great interest the review article entitled ´Comparison of selected characteristics of SARS-CoV3 2, SARS-CoV and HcoV-NL63‘. The topic of the review is highly relevant, the content of the article is logically organized and presented in a clear and readable form, language and style are appropriate. However, the article is superfluous as the matters with which it deals have already been reviewed in various other articles, even in a more detailed or more precise way. The article brings no new information to the reader as well as no new aspects or perspectives for further research. Although such an article might be interesting for some local readers, it does not meet the criteria for publication in an international scientific journal.
Author Response
Dear Reviewer , 01/04/2021
Thank you for giving us the opportunity to improve and resubmit our manuscript:
Comparison of selected characteristics of SARS-CoV-2, SARS-CoV and HCoV-NL63, Darina Bačenková, Marianna Trebuňová, Tatiana Špakovská and Jozef Živčák
Please find enclosed the revised manuscript.
The manuscript has been revised according to the comments raised by the reviewer to the best of our ability. Changes to the manuscript are underscored through "Track Changes" function in
Microsoft Word. The manuscript was checked by an English editor from the editorial office.
We would like to thank the reviewer for the constructive and competent criticism, and we hope that
our manuscript will be acceptable for publication.
Sincerely, Darina Bačenková
Response to the reviewers Reviewer
Comments to the Author and answer.
I read with great interest the review article entitled ´Comparison of selected characteristics of SARS-CoV3 2, SARS-CoV and HcoV-NL63‘. The topic of the review is highly relevant, the content of the article is logically organized and presented in a clear and readable form, language and style are appropriate. However, the article is superfluous as the matters with which it deals have already been reviewed in various other articles, even in a more detailed or more precise way. The article brings no new information to the reader as well as no new aspects or perspectives for further research. Although such an article might be interesting for some local readers, it does not meet the criteria for publication in an international scientific journal.
Answer: We have added a section on structural and non-structural proteins with literature and pictures to the text, part of HCoV-NL63 and role of integrins. The article has been edited in English.
Round 2
Reviewer 2 Report
1.The work remains poorly written and poorly organized.
For example, in the abstract, it is stated: "Identification of the primary causative agent of the virus is essential for a comprehensive knowledge of SARS-CoV-2". I am not sure what is meant here. Overall, text is disjoint and lacks flow.
2. COVID-19 is introduced twice in the text.
3. Please define "SARS-like virus".
4. I believe the authors should define sgRNA as "subgenomic RNA" not "single guide RNA"
Author Response
Manuscript ID: applsci-1040065
Type of manuscript: Review
Title: Comparison of selected characteristics of SARS-CoV-2, SARS-CoV and HCoV-NL63 Darina Bačenková1, * Marianna Trebuňová1, Tatiana Špakovská2, Marek Schnitzer1 and Lucia Bednarcikova1, and Jozef Živčák1
Dear reviewer,
01/24/2021
Thank you for giving us the opportunity to improve and resubmit our manuscript.
The manuscript has been revised according to the comments raised by the reviewer to the best of our ability. Changes to the manuscript are underscored through "Track Changes" function in Microsoft Word.
We have added better quality pictures to the article. A paragraph on the RAAS system has been added and rewritten in the article. We rewrote the abstract of the article.
Comments and Suggestions for Authors
1.The work remains poorly written and poorly organized.
For example, in the abstract, it is stated: "Identification of the primary causative agent of the virus is essential for a comprehensive knowledge of SARS-CoV-2". I am not sure what is meant here. Overall, text is disjoint and lacks flow.
Answer: We rewrote and corrected the abstract. We have omitted the disputed sentence. We have added a better quality image to the article. A paragraph on the RAAS system has been added and rewritten in the article.
- COVID-19 is introduced twice in the text.
Answer: COVID abbreviation and the whole term is given separately in the abstract and separately in the text . The COVID-19 abbreviation is deleted in Table 1 and in text.
- Please define "SARS-like virus".
Answer: SARS-like viruses are mentioned in several scientific references. This means viruses similar to the SARS-CoV coronavirus, which were identified in 2002 and MERS-CoV in 2012, SARS-CoV-2 in 2019. Coronaviruses, which are the causative agents and pathogens of respiratory diseases.
- Cohen, Jon, and Dennis Normile. "New SARS-like virus in China triggers alarm." (2020): 234-235.
- Nain, Zulkar, et al. "Pathogenetic profiling of COVID-19 and SARS-like viruses." Briefings in bioinformatics (2020).
- I believe the authors should define sgRNA as "subgenomic RNA" not "single guide RNA"
Answer: Corrected subgenomic RNA. The authors added sgRNA in the figure 3.
Reviewer 3 Report
Regarding the content, the resubmitted paper meets better the requirements for publication. However, the manuscript contains a lot of inaccuracies and misleading information, particularly concerning the components of the renin-angiotensin system. For example:
Line 23: “Therapeutic RAAS inhibitors directly affect ACE.” – angiotensin II type I receptor blockers (ARBs), direct renin inhibitors or aldosterone antagonists also belong to the inhibitors of the renin-angiotensin-aldosterone system (RAAS), however, these substances affect ACE rather indirectly than directly.
Lines 24-25: “These viruses bind to the receptor-binding domain in ACE2 almost identically.” - The receptor-binding domain is a part of the the CoVs spike protein and not a structural domain of the ACE2; the sentence is thus misleading.
Lines 704-706: “Although angiotensin II is the primary substrate of ACE2, the enzyme angiotensin I cleaves to angiotensin (1–9) and is catalyzed by cathepsin A”. - The sentence is incomprehensible. It is generally accepted that ACE2 converts angiotensin II to angiotensin 1-7 or angiotensin A to alamandine. Angiotensin I can be converted into angiotensin 1-9 for example by carboxypeptidase A or cathpesin A. The in vivo role of ACE2 in this conversion remains disputable due to the poor kinetics of the reaction.
Lines 1171-2: “Differences in the recognition of the RBD structure on ACE are small between these viruses”. - The authors meant probably ACE2, not ACE. Moreover, RBD is a structural part of the virus and not of the receptor, however, the sentence evokes the opposite.
Lines 1235-7: “The effect of RAAS blockers on ACE2 levels suggests that omission of ACE2 blockers may be detrimental rather than beneficial in patients with lung injury.” – ACE2 blockers are not commonly used in patients, the authors meant probably ACE inhibitors instead of ACE2 blockers.
Furthermore, the effect of ACEIs and ARBs on the SARS-CoV-2 infection has been studied and reviewed much more in detail since the onset of the pandemic. The parts of the manuscript discussing the role of the RAAS and RAAS-inhibiting therapies seem generally to be very superficial, giving the impression that the authors are not experts in this field. To sum up, a substantial and careful revision would be required to make the manuscript suitable for publication.
Author Response
Manuscript ID: applsci-1040065
Type of manuscript: Review
Title: Comparison of selected characteristics of SARS-CoV-2, SARS-CoV and HCoV-NL63 Darina Bačenková1, * Marianna Trebuňová1, Tatiana Špakovská2, Marek Schnitzer1 and Lucia Bednarcikova1, and Jozef Živčák1
Dear reviewer, 01/24/2021
Thank you for giving us the opportunity to improve and resubmit our manuscript:
The manuscript has been revised according to the comments raised by the reviewer to the best of our ability. Changes to the manuscript are underscored through "Track Changes" function in Microsoft Word.
We have added better quality pictures to the article. A paragraph on the RAAS system has been added and rewritten in the article.
Comments and Suggestions for Authors
Regarding the content, the resubmitted paper meets better the requirements for publication. However, the manuscript contains a lot of inaccuracies and misleading information, particularly concerning the components of the renin-angiotensin system. For example:
Line 23: “Therapeutic RAAS inhibitors directly affect ACE.” – angiotensin II type I receptor blockers (ARBs), direct renin inhibitors or aldosterone antagonists also belong to the inhibitors of the renin-angiotensin-aldosterone system (RAAS), however, these substances affect ACE rather indirectly than directly.
Answer: I fixed it. ACE2 is well-known as a counter-regulator of the renin-angiotensin system (RAAS) and plays a key role in the cardiovascular system.
Lines 24-25: “These viruses bind to the receptor-binding domain in ACE2 almost identically.” - The receptor-binding domain is a part of the the CoVs spike protein and not a structural domain of the ACE2; the sentence is thus misleading. We rewrote the complete abstract of the article
Answer: I fixed it in abstract. We deleted the inaccurate sentence as recommended. We rewrote the abstract of the article.
The angiotensin converting enzyme-2 (ACE-2) has been identified as the receptor for the SARS-CoV-2 viral entry. ACE2 is well-known as a counter-regulator of the renin-angiotensin system (RAAS) and plays a key role in the cardiovascular system. In therapy of patients with COVID-19, there has been a concern about the use of RAAS inhibitors. As a result, it is hypothesized that ACE inhibitors do not directly affect ACE2 activity in clinical use.
Lines 704-706: “Although angiotensin II is the primary substrate of ACE2, the enzyme angiotensin I cleaves to angiotensin (1–9) and is catalyzed by cathepsin A”. - The sentence is incomprehensible. It is generally accepted that ACE2 converts angiotensin II to angiotensin 1-7 or angiotensin A to alamandine. Angiotensin I can be converted into angiotensin 1-9 for example by carboxypeptidase A or cathpesin A. The in vivo role of ACE2 in this conversion remains disputable due to the poor kinetics of the reaction.
Answer:
We rewrote and supplemented the text in section RAAS Complex. We have added the current citation.
[42] Bian, J.; Li, Z. Angiotensin-converting enzyme 2 (ACE2): SARS-CoV-2 receptor and RAS modulator. Acta Pharmaceutica Sinica B, 2021, 11, 1-12
[71] de la Cruz, A.; Ashraf, S.; Vittorio, T.J.; Bella, J.N. COVID-19 and renin-angiotensin system modulators: what do we know so far? Expert review of cardiovascular therapy, 2020, 18, 743-748.
Lines 1171-2: “Differences in the recognition of the RBD structure on ACE are small between these viruses”. -
Answer: I fixed it. We have added the current citation and text. We deleted the inaccurate sentence as recommended.
[71] Tang, T.; Bidon, M.; Jaimes, J.A. ; Whittaker, G. R. ; Daniel, S. Coronavirus membrane fusion mechanism offers a potential target for antiviral development. Antiviral research, 2020, 178, 104792.
Lines 1235-7: “The effect of RAAS blockers on ACE2 levels suggests that omission of ACE2 blockers may be detrimental rather than beneficial in patients with lung injury.” – ACE2 blockers are not commonly used in patients, the authors meant probably ACE inhibitors instead of ACE2 blockers.
Answer: I fixed it. Corrected: ACE
Round 3
Reviewer 3 Report
I have read the revised version of the manuscript. I accept the responses and the changes introduced to the paper and recommend it for publication.